# Protein Binding to Cis-Motifs in mRNAs Coding Sequence Is Common and Regulates Transcript Stability and the Rate of Translation

**DOI:** 10.3390/cells10112910

**Published:** 2021-10-27

**Authors:** Ewa A. Grzybowska, Maciej Wakula

**Affiliations:** Molecular and Translational Oncology, Maria Sklodowska-Curie National Research Institute of Oncology, Roentgena 5, 02-781 Warsaw, Poland; maciej.wakula@pib-nio.pl

**Keywords:** RNA-binding proteins, coding sequence, mammalian post-transcriptional gene expression

## Abstract

Protein binding to the non-coding regions of mRNAs is relatively well characterized and its functionality has been described in many examples. New results obtained by high-throughput methods indicate that binding to the coding sequence (CDS) by RNA-binding proteins is also quite common, but the functions thereof are more obscure. As described in this review, CDS binding has a role in the regulation of mRNA stability, but it has also a more intriguing role in the regulation of translational efficiency. Global approaches, which suggest the significance of CDS binding along with specific examples of CDS-binding RBPs and their modes of action, are outlined here, pointing to the existence of a relatively less-known regulatory network controlling mRNA stability and translation on yet another level.

## 1. Introduction

RNA-binding proteins (RBPs) regulate mRNA expression on many levels. Usually, they are associated with 5′UTR or 3′UTR-binding; these non-coding regions have a long-established regulatory role and have been demonstrated to regulate pre-mRNA splicing, cleavage and polyadenylation, RNA stability, RNA localization, RNA editing, and translation. However, there is mounting evidence that protein-coding regions are also targeted by some RBPs. What is the role of this binding?

As observed by Lee and Gorospe [1] the coding region was for long neglected as a possible source of post-transcriptional coding, but this view is starting to change. While 5′UTR has been linked to the regulation of translation initiation and there are numerous examples of it [2], 3′UTR is widely considered as the hub of mRNA stability regulation [3]. Numerous proteins, as well as miRNAs, bind to this region, promoting the stabilization or degradation of transcripts. Also, 3′UTR has been implicated in the regulation of mRNA localization, translation, protein conformation, and protein-complex formation, as well as in some post-translational modifications [3]. The mRNA coding sequence was, however, considered non-regulatory and was supposed not to interact with proteins. New data challenge this view, and, although—For the moment—There are not many examples of the resolved regulatory mechanisms of the CDS-binding, the list is growing. In this review we aim to characterize the existing evidence of protein binding to CDS, describe known examples of the role of this binding in the cell and propose some functional possibilities.

## 2. Global Approaches to Characterizing RBP-Binding Sites Suggest a Significant Subset of Targets in CDS

The development of high-throughput techniques, which enable the detection of RBP-binding sites on the level of the whole transcriptome and to pinpoint the exact binding sequence, vastly expanded the catalogue of the known functional elements encoded in the human genome (Box 1). Moreover, recent mass spectrometry-based methods have identified hundreds of previously unknown proteins bound to RNA (for example, according to the current estimates, the human genome may contain about 1500 or more RBP-encoding genes [4]). In effect, we have obtained a huge amount of new data, to which we need to fit, in a whole, regulatory sense.

New data from a large-scale (transcriptome-wide) mapping of RBP binding sites provide, somewhat surprisingly, the evidence for RBPs targeting coding sequence regions.

Yang et al. [5], assembled a database (CLIPdb) of RBP-RNA interactions based on 395 publicly available CLIP-seq data sets for 111 RBPs from four organisms (human, mouse, worm and yeast). Their analysis of genomic elements identified as RBD targets revealed a substantial proportion of CDS target sites (14.2% in human, 16.8% in mouse)-numbers comparable to 3′UTR occupancy (24.6% and 19.3% in human and mouse, respectively). Interestingly, they also report quite high proportion of intronic regions (42.9–51.2%). This high proportion of intronic sequences is especially characteristic in the case of splicing factors (for example SRSF3, SRSF4 [6], Tra2β [7]). However, since introns are much longer than exons, these comparisons should take into account the difference in length and normalize for it (tag-density calculations).

Van Nostrand et al. [8], describe the characterization, using integrative approaches, of 356 human RBPs and the in-vivo binding eCLIP assays of 150 RBPs, introducing a new dataset of RNA elements that are recognized by RNA-binding proteins. These binding assays indicate that for many RBPs, mRNA coding sequence constitutes a large portion of all characterized RNA elements. As many as 22.4% of all analyzed RBPs had more than 50% binding sites located in CDS, for 35% RBPs CDS targeting was between 1–50% and only 1.8% of RBPs did not have any CDS-located binding sites. For some RBPs, CDS-binding was prevalent, for example, for FXR1, FXR2 and RPS3 it was more than 90%, while, for TRA2A, it was around 89%.

Srivastava et al. [9] characterize RNA targets of 97 RBPs using Protein Occupancy Profile-Sequencing (POP-seq), which is reported to be less biased than other high-throughput methods. They have demonstrated that a majority of the peaks indicating RBP-binding in K562 cells were mapped to exons (~48%), while a relatively lower proportion were mapped to introns (~19%) and 3′UTR’s (~16%).

These novel results indicate that CDS binding is not incidental and requires extensive explanation.

## 3. Evolutionary Constraints to CDS Binding

A primary constraint in coding sequence evolution is the need to preserve encoded protein function, thus sequences encoding any sense (protein-encoding or regulatory) are always more conserved, and this universal rule enables molecular clocks to tick. However, any single sequence can contain co-existing multiple layers of information, and in case of CDS, in addition to the obvious message encoded in a sequence, the other layers include specific structural elements and RNA-binding motifs, enabling binding of specific regulatory RBPs. This multicoding implies preservation of not only regions important for protein function, but also of RBP-binding motifs, whether structural or linear, with (presumably) regulatory functions.

Ramakrishnan and Janga [10] analyzed the extent of conservation of binding sites for 60 human RBPs (CLIPdb data) and concluded that, while generally highly conserved, their level of conservation is related to expression level and number of targets of an RBP. They also concluded that binding sites occurring on 3′ ends are the most conserved and, although this was not discussed, they provide a heatmap suggesting that binding sites occurring in CDS are comparably conserved.

Savisaar et al. [11] pointed out that the selection pressure exerted on the CDSs is directed to preserving RBP-binding sites, but, simultaneously, the evolution of these regions is also constrained by the need to avoid unwanted RBP-binding, which might be detrimental for the mRNA. The authors analyzed data from several databases and compiled sequence specificities for 114 RBPs, resulting in a final list of 1483 unique k-mers. They concluded that RBP motif-related constraint, calculated using motif density and conservation parameters for various regions, is as strong in CDSs as in non-coding regions of mRNAs. They also estimated that conservation of RBPs-binding sites leads to a decrease of 2−3% in the overall rate of evolution at synonymous sites, which is comparable to the range of a decrease (1.9–4%) obtained for exonic splice enhancers (ESEs).

## 4. Structural Elements in Coding Sequence Are Conserved and Bind RBPs

RNA can adopt complex secondary structures, which constitute protein-binding, regulatory motifs. These elements are evolutionarily conserved, and the structure may be even more conserved than the sequence. Casas-Vila et al. [12] applied RNA-centric method (quantitative RNA pull-down) coupled with mass spectrometry to analyze RBPs binding to 186 evolutionary conserved RNA structures in *S. cerevisiae*. They have found 162 interacting proteins. Detailed analysis of positional binding preferences of these RBPs revealed that, although 25% acted irrespective of the functional region of the mRNA, 50% of them showed such preferences, with a significant number binding to RNA folds in the coding regions of mRNAs. GO-term enrichment analysis ranked by genomic position of the fold (5′UTR, CDS or 3′UTR) showed enrichment of GTPase activity acting to initiate translation for the 5′UTR-binding RBPs, the structural constituents of the ribosome and helicase activity for CDS, and the nuclease involved in mRNA catabolism for 3′UTR.

Additionally, it was shown that, in spite of the potential role of structural RNA elements in protein binding, RBPs preferentially bind low-complexity motifs [13], often present in disordered proteins. They can also bind interspersed, bi- or tri-partite short motifs in RNA, flanking structural elements. Such a mode of binding and, generally, of context-specific binding is, of course, more difficult to study in a comprehensive, high-throughput manner.

## 5. Functionality of CDS Binding

### 5.1. mRNA Stability Regulation Is Coupled with Translation

The regulation of mRNA stability is a well-known function of RBPs, however, it was usually attributed to 3′UTR binding sites [14]. Numerous mechanisms have been described, explaining either degradation or stabilization of the transcript mediated via 3′UTR binding, but also localization of mRNA, translation and co-translational complex formation [14,15]. The most known is probably RBP binding to AU-rich elements (ARE), originally described as involved in a rapid degradation of specific transcripts [16], but later found to be also involved in transport and translation [17]. Except ARE-binding, there are numerous other examples of destabilization, stabilization or translation regulation driven by RBPs binding to this region [18]. 3′UTR is also the location of miRNA binding and its powerful regulatory potential is well established. Also, 3′UTR is ideal for such regulation, since its RBP binding sites are not constrained by the need for preservation of the encoded message as in CDS or by the direct involvement in translation initiation, as in 5′UTRs.

The main confusion surrounding the issue of stability regulation via CDS binding, lies in a fact that in normal circumstances CDS should be covered with ribosomes, hence it should be both, well protected from degradation and difficult to access by RBPs. RBPs sitting on the coding sequence would also pose a roadblock to the translating ribosomes, disrupting the process. Thus, logic dictates that RBP-binding to CDS should pertain to a situation when mRNA is not actively translated, as during storage in RNA granules or during cellular transit to the site of localized translation. Alternatively, it may be a method of slowing down translation. Several possibilities of CDS-binding functionality (as discussed in details below) are described in Figure 1.

#### 5.1.1. CRD-BP

The described examples of stability regulation achieved via CDS binding are so far sparse and seem to be well-connected to the regulation of translation, which is in line with the above reasoning. In fact, dual regulatory effects of the same RBPs, impacting both mRNA stability and translation rates, have been postulated for some RBPs [19]. The two functionalities seem to be intertwined also in the case of CDS-binding RBPs. One example is coding region determinant-binding protein (CRD-BP), a mouse protein related to chicken ZBP1 (zipcode-binding protein 1) and human IMP-1,2,3 (insulin-like growth factor 2 mRNA-binding protein, IGF2BP). A 249-nucleotide region, termed coding region instability determinant, was shown to destabilize c-myc mRNA (Figure 1B) [20]. In the follow-up research [21], it was shown that mRNA degradation is caused by endonuclease attack, possibly because of the translational pausing on the CRD, resulting in ribosome-free part of the transcript downstream of the pausing site, which expose it to endonuclease cleavage. CRD-BP binding protects and stabilizes c-myc mRNA. Translational pause is caused by the presence of rare codons, arginine (CGA) and threonine (ACA), so the process stops due to the lack of the appropriate tRNA. The authors have demonstrated that changing those codons to more common ones increases transcript abundance.

#### 5.1.2. GLD-1

Another example of CDS-binding stabilization is GLD-1, the translational regulator of *Caenorhabditis elegans* [22]. This RBP contains hnRNP K homology (KH) RNA-binding domain and is a member of STAR family (signal transducer and activator of RNA metabolism). Again, the dual role of GLD-1 was postulated: in transcript stabilization and translation repression. Brümmer et al. [22], reported their modelling of the binding specificity of GLD-1, based on the five previously established PAR-CLIP libraries (iPAR-CLIP, HITS-CLIP). Using these data, they have established a set of GLD-1 binding motifs and performed transcriptome-wide prediction of GLD-1 binding. Since they expanded the primary library data obtained by Jungkamp et al. [23], they were able to characterize more CDS-binding sites than previously described and, in fact, their model predicts more binding sites in CDS than in 3′UTR and 5′UTR. Additionally, they have analyzed the functional effects of GLD-1 binding depending on the genomic position of the binding site and concluded that mRNA stabilization requires interaction with high-affinity sites within 3′UTRs, while translational repression is associated with binding sites in the CDS that appear to have lower affinity [24]. On the other hand, Theil et al. [25] argue that translational repression is mediated mostly by high-affinity 5′UTR-binding, not CDS low-affinity binding. However, it must be noted, that low-affinity binding can be quite effective, if compensated for by the number of sites—which is often the case with unstructured RBPs binding with low-complexity motifs.

#### 5.1.3. UNR

Another CDS-binding RBP, Upstream-of-N-Ras protein (UNR, CSDE1 in mammals) has five cold shock domains that bind single-stranded RNA [26]. Drosophila UNR was shown by RIP to bind to hundreds of transcripts [27]. These results were confirmed and expanded upon by, Wurth et al. [26], with iCLIP experiments on melanoma cells. The authors also demonstrated that UNR/CSDE1 regulates a set of transcripts involved in invasion and metastasis in melanoma.

UNR binds mature mRNA in CDS, 3′UTR and, to less extent, 5′UTR regions [26]. Different modes of binding to the coding and non-coding regions, as well as different functionalities of these binding, have been observed for UNR. Typical ofr CDS-binding RBPs, UNR has multiple functions in mRNA regulation, which include: (1) the destabilization of transcripts by binding to their CDS, for example, in the case of c-fos mRNA, where destabilization of mRNA occurs in a translation-dependent manner [28] (Figure 1C); (2) the regulation of internal ribosome entry site (IRES)-dependent translation of the transcripts encoding c-myc, the cell-cycle kinase PITSLRE, and the apoptosis regulator APAF-1, as well as its own IRES [29,30,31,32]; and (3) the regulation of translation of its melanoma targets, mainly at the levels of elongation or termination [26].

### 5.2. Ribosome Stalling and Its Role in Regulating Translation

Ribosome stalling seems to be an important element of translation regulation by CDS-binding RBPs. It appears to be a natural consequence of protein binding to the region when active translation is supposed to take place, but not many such examples have been comprehensively described. In general, ribosome stalling may be mediated by several other mechanisms: (1) by rare codons in messenger RNA, resulting in the lack of the appropriate tRNA, as is described in the case of CRD-BP; (2) by unfavorable amino acid pairing, resulting in slow peptidyl-transfer kinetics (Pro–Pro); (3) by positively charged residues or non-optimal codon clusters in the nascent peptide interacting with the negatively charged ribosome exit tunnel [33,34]; and (4) by physical obstruction mediated by some strong structural element in mRNA. When stalling is permanent, it needs to be resolved by a dissociation of ribosomal subunits, subsequent degradation of the associated mRNA and the nascent polypeptide, and the return of the ribosomal subunits (especially limiting 40S subunit) to the pool of translationally available ribosomal components. This process of ribosome rescue is mediated by the Dom34-Hbs1 complex (PELO-HBS1 in humans) [35]. However, since translational pausing is quite common, and has, in fact, a physiological function—it is usually resolved in a productive manner, for example when strong secondary structures (stem loops and pseudoknots) mediate programmed ribosomal frameshift, or when translation, stalled due to specific amino acid combinations, is resumed with help of auxiliary factors like eIF5A [36]. It can be also a regulatory pause caused by RBP bound to CDS, as in the case of FMRP, described below.

#### 5.2.1. FMRP

Synaptic regulator FMRP (fragile X mental retardation protein encoded by FMR1 gene) binds to the coding sequences of some specific transcripts, creating a substantial roadblock for efficient translation [37,38]. Thus, the depletion of FMRP, which results in Fragile X syndrome, causes harmful increase in protein synthesis, which is linked to synaptic and behavioral defects and neurodegenerative phenotype in Fragile X-associated tremor/ataxia syndrome [39].

Translational repression mediated by FMRP is very complex and encompasses more than one mechanism. Apart from ribosome stalling by CDS binding it also includes forming a complex with CYFIP (Cytoplasmic FMRP Interacting Protein), a protein which interacts with eIF4E on the 5′UTR of transcripts, thus interfering with translation initiation [40]. Another proposed mechanism includes inhibition of translation elongation by recruiting microRNAs, Argonaute 2 (AGO2), and Dicer to form the RNA-induced silencing complex [41,42]. FMRP was also shown to bind directly to the ribosomal subunits, which provokes the question of whether its impact on translation might be, at least in part, mRNA-independent.

Overall, sixty-six percent of FMRP mRNA binding was mapped to the coding region, with no specific position relative to the start and stop codons [38]. FMRP binds directly to G-quartet structures in 3′UTR [43,44], but it was also shown to bind short RNA motifs [45] or bind non-specifically to the CDS [38]. These different modes of binding may correspond to different functions; for example, G-quartet binding was linked to mRNA localization in neurons [46]. However, it was also reported that FMRP binds to CDS at a guanine-rich, G-quartet-like sequence [47].

FMRP regulates a set of plasticity-related proteins that constitute 30% of the postsynaptic density (PSD) proteome. Its targets are numerous and include: PSD-95, Arc, Shank1, NMDA (N-methyl-D-aspartate) receptor subunits, the metabotropic glutamate receptor 5 (mGluR5), CaMKIIα10, striatal-enriched protein tyrosine phosphatase (STEP), the potassium channels Kv3.1 and Kv4.2, and the upstream regulator of phosphoinositide 3 kinase (PI3K), PI3K enhancer (PIKE) [48,49]. Most of the targets represent proteins regulating neuronal development and synaptic structure and function.

FMRP-mediated stalling has been linked to synaptic function and autism [38]. Stalling was demonstrated to be reversible and transcript-specific; the loss of FMRP was shown to relieve ribosome stalling, and it was also shown that FRMP stalls elongating ribosomes only in its phosphorylated form [38,50]. All this suggests the regulatory function of stalling (Figure 1D).

As mentioned before, FRMP binds to mGluR5 mRNA; it also binds competitively to APP (amyloid precursor protein) mRNA and represses translation in response to metabotropic glutamate receptor agonist DHPG (dihydroxyphenylglycol) [47]. FMRP binds to the same coding region element in APP mRNA as another RNA-binding protein, heterogeneous nuclear ribonucleoprotein (hnRNP) C. Both proteins regulate APP translation competitively and in opposite directions [51]. It has been proposed, that FMRP represses translation by recruiting APP mRNA to processing bodies (P-bodies), where it cannot be accessed by the ribosome. Competitive hnRNP C binding restores APP translation by preventing its storage in P-bodies [51]. Thus, this represents yet another form of FMRP interfering with translation.

#### 5.2.2. m6A and YTHDC2

Another example of regulating translation efficiency by ribosome pausing is the methylation of N6 adenosine. N6-methyladenosine (m6A) modification is present in mature mRNAs, with about 35% in the coding sequence. It has been demonstrated that the presence of m6A in the CDS delays translation elongation [52], but the removal of CDS m6A from methylated transcripts did not result in higher translation efficiency, quite opposite, further reduced translation [53]. This counter-intuitive result was explained by the presence of secondary structures in methylated mRNAs; methylation recruits to this region an RBP YTHDC2, RNA helicase-containing m6A reader protein, which resolves these structures, facilitating translation. Thus, m6A acts differently in highly structured transcripts, promoting translation by resolving structural roadblocks, but, for poorly structured mRNAs with active translation, it reduces elongation efficiency [53].

### 5.3. Timing Translation Is Important for Co-Translational Protein Folding and Co-Translational Complex Formation

What could be the physiological role of ribosome stalling? Ribosome pausing prolongs the time required to complete translation in a regulated manner; this timing may have a role in ensuring the quality of the protein product. It was shown, for example, that the local slowing of translation by non-optimal codons is needed for co-translational recognition of the nascent peptide by the signal recognition particle (SRP) [34]. SRP binding allows delivery of a nascent protein to the endoplasmic reticulum, through a translocon channel in the ER membrane. This is the most common for secretory pathway proteins, but SRP-dependent translation was also demonstrated for cytoplasmic proteins [54]. Co-translational targeting to the membrane is the best way to protect these proteins from aggregation in the cytoplasm. Moreover, proper folding of the nascent polypeptide requires a set of the appropriate chaperones, present in the ER lumen, but not in the cytoplasm. This is an example of an elongation-slowing mechanism controlling the proper folding of the protein product. It is easy to imagine that the other mechanisms, including RBP-binding to the CDS, may serve the same purpose.

Different kinetics of translation and co-translational folding may result in the increase of misfolded protein products and subsequent degradation. Altered protein conformation caused by aberrant folding may also result in different oxidation states and different functionalities (or lack thereof) [55].

Protein folding is much faster than the slow growth of the polypeptide chain, but slowing the rate of translation at the critical points has been proven to affect protein stability, solubility, and conformation [56]. Clusters of rare codons are observed in specific regions; for example, they are often found in the first 90–100 nucleotides and on the domain boundaries, serving as timers that ensure vectorial folding and prevent inter-domain aggregation [57,58]. It was demonstrated for some proteins (*Neurospora crassa* FREQUENCY, mammalian gamma B-crystallin) that optimization of codons leads to a higher protein production, but the functionality of these proteins is disrupted, due to their altered structure [59,60].

Timing translation is even more important for co-translational complex formation, which requires the concerted action of many factors. A study conducted on yeast revealed that co-translational subunit association is a prevalent mechanism for complex assembly [61], and it is highly probable that the same holds true for all Eukaryotes. A co-translational complex assembly is generally uni-directional, with subsequent subunits engaging with the previous fully synthesized and folded subunits, thus counteracting the propensity for aggregation of the nascent subunit [61].

While there are no examples or direct evidence of the involvement of CDS-binding RBPs in timing co-translational complex assembly, it seems probable that the translational pause induced by RBP-binding may provide time necessary to the proper folding of subsequent subunits, as well as for the recruitment of the required factors (components of the complex and the appropriate chaperones) to the translation site. Further research should prove or disprove this possibility.

## 6. Conclusions

While there are still not many examples of satisfactorily resolved functionality in RBP’s binding to CDS, some patterns seem to emerge. Firstly, as evidenced by high-throughput data, RBP’s binding to CDS appears to be much more common than previously expected. Secondly, in all the described examples, functions in the regulation of mRNA stability and translation (initiation or elongation) are intertwined, sometimes in a very complex way. There are also many examples of dual-, multi-, or context-dependent functionality. CDS binding seem to be an element of the complex fine-tuning of expression, which is especially important in more complex Eukaryotes.

It should be noted that not all known interactions or possible functions were described in this review; for example, two splicing factors, SRSF3 and SRSF4, were also shown to bind to the coding sequence of some transcripts [6], but the functional significance of this binding is unclear at the moment. Similarly, it was demonstrated that the splicing activator, Tra2β, binds frequently to exonic sequences, which are evolutionarily conserved [7]. These examples clearly suggest the role of CDS binding in the regulation of splicing or alternative splicing.

There are still many paths to explore; for example, while not evidenced, it seems plausible that CDS binding may function as a timer of translation and, as such, it may regulate proper folding. Another not-fully-explored possibility lies in the connection of CDS-binding-mediated translational pause to situations when active translation should be prevented, as during mRNA storage or its transport to correct localization in the cell. Overall, while, for the moment, the main functions of CDS-binding seem to include transcript stabilization or translation repression, further research should bring more complex, precise, and intricate explanations.

Box 1A short guide to the high-throughput methods used for the characterization of protein–RNA interactions.**RIP**—RNA immunoprecipitation: Simple immunoprecipitation with RBPs of interest, cells unfixed or after formaldehyde fixation. Antibody-RNP immunocomplexes are purified on aga rose or magnetic beads and the RBP of interest is removed by proteinase K. After extraction, RNA is transcribed to cDNA, which might be analyzed via qPCR, microarrays (RIP-chip) or next generation sequencing (RIP-seq).**CLIP**—(UV) Crosslinking and immunoprecipitation: Similar to RIP, but cell lysis is preceded by UV-C crosslinking, which, in contrast to formaldehyde, preserves only protein–RNA interactions. This step allows the application of restrictive purification conditions, even in cases of weak RNA–RBP interactions, decreasing the chance of sample contamination by non-specific RBPs. The efficiency and specificity of isolating RNA targeted by RBPs of interest is higher when compared with RIP. Specificity is further increased by the fact that both protein–RNA complexes and cDNA are separated by electrophoresis and extracted from the membrane and gel, respectively.**PAR-CLIP**—Photoactivable ribonucleoside-enhanced CLIP: In contrast to the basic version, in this CLIP variation the first step consists of incubation with photoactivatable ribonucleosides (4-thiouridine or 6-thioguanosine), the incorporation of which into RNA allows the use of UV-A crosslinking and, as a consequence, alters the spectrum of types of covalent bounds generated and protein–RNA interactions to be detected.**CRAC**—(UV) crosslinking and analysis of cDNAs: The quality of immunoprecipitation of protein-RNA complexes depends on the specificity and affinity of the antibody to the RBP, such that the quality of the antibody may be the limiting factor. To improve specificity, this variation of CLIP involves a two-phase purification step (e.g., utilizing molecular tags like protein A or his-tag).**iCLIP**—Individual-nucleotide resolution CLIP: Each type of the described CLIP methods involves adaptor ligation, followed by reverse transcription and later cDNA amplification. This process is inefficient in the case of nucleotides that have been previously crosslinked to the RBP, resulting in premature termination, which generates a significant number of truncated DNA fragments. To overcome this issue, during iCLIP, the primer complementary to the 3′ adapter contains molecular the barcode and sequence of the second primer. After reverse transcription, fragments are circularized and then linearized. Consequently, the RBP target sequence is flanked by the primer sequences at both the 3′ and 5′ end. This method allows not only the preservation of these fragments, but also, due to the barcode presence, the identification of the position of the crosslinked nucleotides.**eCLIP**—Enhanced CLIP: This method utilizes a two-phase adapters step to increase the efficiency of the generation and amplification of purified cDNA in CLIP experiments. First ligation of the RNA with an indexed 3′ RNA adapter occurs when the RNP is attached to beads during immunoprecipitation. After purification, elution and reverse transcription of the 3′ ssDNA adapter is ligated to the cDNA fragment, allowing single-nucleotide resolution of iCLIP.**POP-seq**—Protein Occupancy Profile-sequencing: This technique is based on a multi-step phase separation with a trizol reagent, without the requirement of poly-A pulldown. Protein-RNA complexes accumulate in the interphase between the aqueous phase and the organic phase and are isolated by the three subsequent cycles of phase separation. The purified target RNA is used to generate next-generation sequencing libraries.

## Figures and Tables

**Figure 1 cells-10-02910-f001:**
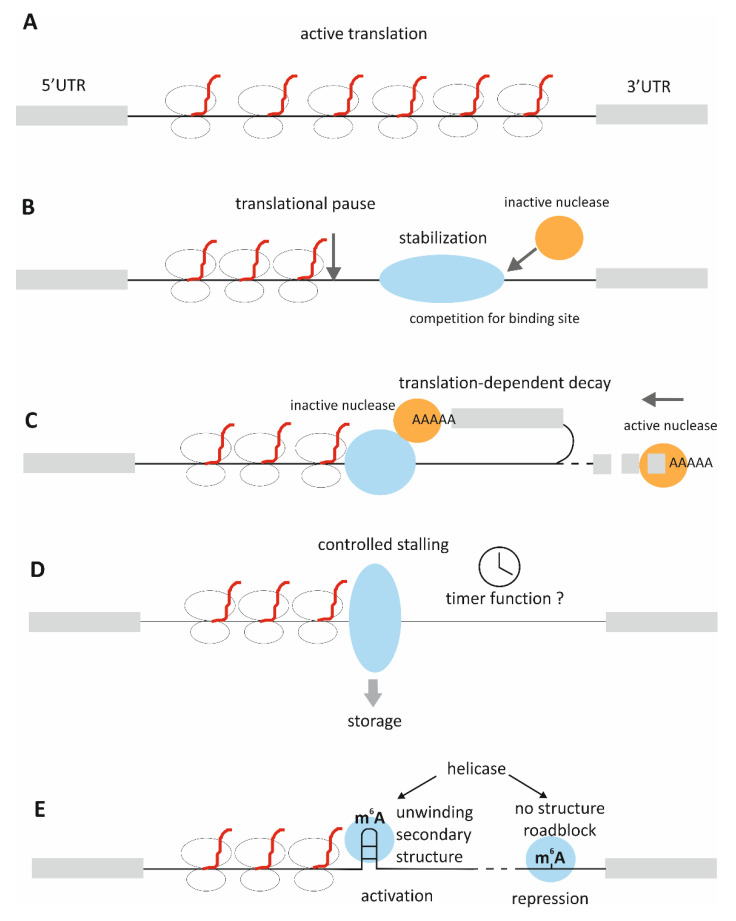
Proposed functions of CDS-binding by RBPs. (**A**). Active translation—CDS is uniformly covered by ribosomes and protected from degradation. Translation is regulated mostly by cis-elements present in 5′UTR or 3′UTR (**B**). Translational pause introduced by rare or suboptimal codons or secondary structure in mRNA stalls ribosomes, leaving the rest of the transcript uncovered and unprotected; RBP-binding stabilizes the transcript, competing for the binding site with endonuclease, as in case of CRD-BP; translation is slower but the transcript is preserved (**C**). CDS-binding RBP interacts with polyA-binding nuclease, which stabilizes the transcript until translation reaches this region, causing disruption of the complex and subsequent degradation of the mRNA by the activated nuclease; as in case of UNR and c-fos mRNA (**D**). RBP-binding stalls translation in a controlled manner, for example regulated by phosphorylation; may be linked to storage in P-bodies; as in case of FRMP (**E**). Secondary structure in mRNA stalls translation; its methylation recruits a helicase, which unwinds the structure and relieves translation; when the helicase binds to the unstructured part it blocks translation, as in case of m6A and YTHDC2.

## Data Availability

Not applicable.

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
