# Peer review of "Protein Binding to Cis-Motifs in mRNAs Coding Sequence Is Common and Regulates Transcript Stability and the Rate of Translation"

_cells, 2021, doi:10.3390/cells10112910_

Round 1

Reviewer 1 Report

This is a literature review on the function of RNA-binding proteins that bind to CDS. I recommend publishing this review because it is well-written and clearly structured.

No major revisions are needed.
Here are some minor points.

1) It is recommended that the authors indicate the Figure number in the text. For example, "Figure 1D" would be inserted somewhere in the FMRP section.
2) There are double spaces in the text (lines 28,30,32,173,188,211,239).

Author Response

Additional references to Figure 1 have been inserted in several places in the revised version of the manuscript. Double spaces have been deleted.

Reviewer 2 Report

In this review, the authors describe recent advances in the field of RNA binding proteins (RBP) recognizing sequences present within the mRNA coding sequence. Indeed, recent studies have revealed that RBP not only interact with specific signature present in the 5´ or 3´ UTR but are also recruited to the CDS. Using few well selected examples among the most characterized proteins, the authors nicely summarize our current knowledge in this field, showing for instance that RBPs can modulate translation speed to allow some proteins domains to fold or can stabilize mRNAs from the degradation by endonucleases. Most probably other mechanisms exist but as they are not highly documented, the authors prefer not to discussion those cases.

Overall, this review is well written and of evident interest. It contains a box, which very nicely described the most popular techniques available to identify the set of proteins bound to mRNAs and to obtain information at the nucleotide regarding their binding sites. 

There are few typos listed below that should be corrected before publication. 

  • Lane 102 : "Over-all" should be "overall".
  • Lane 174 : "then" should be "than".
  • Lane 176 : remove one duplication of "that"
  • Lane 477 : The format of this reference should be modified as names are in capital letters.
  • Lane 543 : For reference 59, only initials of the authors are indicated. This should be corrected.

Author Response

Spelling mistakes and the format of references have been corrected.